# Unsupervised Hierarchical Clustering of Pancreatic Adenocarcinoma Dataset from TCGA Defines a Mucin Expression Profile that Impacts Overall Survival

**DOI:** 10.3390/cancers12113309

**Published:** 2020-11-09

**Authors:** Nicolas Jonckheere, Julie Auwercx, Elsa Hadj Bachir, Lucie Coppin, Nihad Boukrout, Audrey Vincent, Bernadette Neve, Mathieu Gautier, Victor Treviño, Isabelle Van Seuningen

**Affiliations:** 1Univ. Lille, CNRS, Inserm, CHU Lille, UMR9020-U1277-CANTHER—Cancer Heterogeneity Plasticity and Resistance to Therapies, F-59000 Lille, France; nicolas.jonckheere@inserm.fr (N.J.); julie.auwercx@etud.u-picardie.fr (J.A.); elsa.hadj-bachir@inserm.fr (E.H.B.); lucie.coppin@inserm.fr (L.C.); nihad.boukrout@inserm.fr (N.B.); audrey.vincent@inserm.fr (A.V.); bernadette.neve@inserm.fr (B.N.); 2Laboratoire de Physiologie Cellulaire et Moléculaire-UR UPJV 4667, UFR Sciences, Université de Picardie Jules Verne (UPJV), 80039 Amiens, France; mathieu.gautier@u-picardie.fr; 3Cátedra de Bioinformática, Escuela de Medicina, Tecnologico de Monterrey, Monterrey 64710, Nuevo León, Mexico; vtrevino@itesm.mx

**Keywords:** pancreatic cancer, TCGA, mucin, gene signature, unsupervised hierarchical clustering, overall survival

## Abstract

**Simple Summary:**

Pancreatic cancer has a dramatic outcome (survival curve < 6 months) that is the consequence of late diagnosis and the lack of efficient therapy. We investigated the relationship between the 22 mucin gene expression and the patient survival in pancreatic cancer datasets that provide a comprehensive mapping of transcriptomic alterations occurring during carcinogenesis. Using unsupervised hierarchical clustering analysis of mucin gene expression patterns, we identified two major clusters of patients: atypical mucin signature (#1; MUC15, MUC14/EMCN, and MUC18/MCAM) and membrane-bound mucin signature (#2; MUC1, -4, -16, -17, -20, and -21). The signature #2 is associated with shorter overall survival, suggesting that the pattern of membrane-bound mucin expression could be a new prognostic marker for PDAC patients.

**Abstract:**

Mucins are commonly associated with pancreatic ductal adenocarcinoma (PDAC) that is a deadly disease because of the lack of early diagnosis and efficient therapies. There are 22 mucin genes encoding large *O*-glycoproteins divided into two major subgroups: membrane-bound and secreted mucins. We investigated mucin expression and their impact on patient survival in the PDAC dataset from The Cancer Genome Atlas (PAAD-TCGA). We observed a statistically significant increased messenger RNA (mRNA) relative level of most of the membrane-bound mucins (*MUC1/3A/4/12/13/16/17/20*), secreted mucins (*MUC5AC/5B*), and atypical mucins (*MUC14/18*) compared to normal pancreas. We show that *MUC1/4/5B/14/17/20/21* mRNA levels are associated with poorer survival in the high-expression group compared to the low-expression group. Using unsupervised clustering analysis of mucin gene expression patterns, we identified two major clusters of patients. Cluster #1 harbors a higher expression of *MUC15* and atypical *MUC14*/*MUC18*, whereas cluster #2 is characterized by a global overexpression of membrane-bound mucins (*MUC1/4/16/17/20/21*). Cluster #2 is associated with shorter overall survival. The patient stratification appears to be independent of usual clinical features (tumor stage, differentiation grade, lymph node invasion) suggesting that the pattern of membrane-bound mucin expression could be a new prognostic marker for PDAC patients.

## 1. Introduction

Pancreatic cancer, with its major form pancreatic ductal adenocarcinoma (PDAC), is projected to become the second cause of death by cancer and the first among digestive cancer worldwide by 2030 [1]. The short survival curve (6 months) and low 5 year survival rate (9%) is the consequence of late diagnosis and the lack of efficient therapy [2,3]. Interpatient heterogeneity and phenotypic differences between PDAC were reported and stratified as basal or quasi mesenchymal [4,5,6,7]. The daily clinical impact of this classification remains to be fully demonstrated, but the obvious differences regarding patient survival suggest the possibility to use prognostic expression signatures and may determine the relationship between signature and efficacy of therapies [8]. The Cancer Genome Atlas (TCGA) provides a comprehensive mapping of the key genomic changes, transcriptomic and proteomic alterations occurring during carcinogenesis. Notably, the pancreatic adenocarcinoma (PAAD) dataset contains DNA alterations or methylation, and messenger RNA (mRNA), microRNA (miRNA), long noncoding RNA (lncRNA), and protein expression profiles of 184 patients (including 168 with available mRNA data) that can be used to investigate the involvement of genes of interest [9].

Mucins are large *O*-glycoproteins that are classified into two subgroups: (i) secreted mucins (MUC2, -5AC, -5B, -6, -7, -8, -9/OVGP1 and -19), which are the main components of mucus covering the different epithelia [10,11,12,13], and (ii) membrane-bound or transmembrane mucins (MUC1, -3A/B, -4, -12, -13, -15, -16, -17, -20, -21, and -22), which are bound to cell surface and are involved in interactions with the microenvironment and regulate cell signaling [11,14]. Most of the cytoplasmic tails of membrane-bound mucins can be phosphorylated and, thus, regulate signaling pathways involved in cell–cell interactions, differentiation, and apoptosis [15].

Among the best-known membrane-bound mucins, MUC1 and MUC4 have been extensively proposed as drivers of pancreatic carcinogenesis because they promote tumor growth, proliferation, oncogenic signaling, cell metabolism, epithelial–mesenchymal transition (EMT), and metastasis [14,16,17]. MUC1 is overexpressed in pancreatic cancer. MUC4, as well as MUC5AC and MUC16, is neoexpressed in early neoplastic lesions (PanINs) that progress toward adenocarcinoma [16,17]. MUC16, best known as the ovarian cancer biomarker CA125, also promotes cell proliferation, alters Focal adhesion kinase (FAK)/ Mitogen-Activated Protein Kinases (MAPK) signaling, and facilitates metastasis in pancreatic cancer cells [18,19,20]. Other variants of mucin glycosylated forms such as CA15-3 or CA19.9 are potential PDAC biomarkers but with a limited specificity or sensitivity balance [21,22]. Circulating MUC5AC has also been proposed as a potential biomarker, alone or in combination with CA19-9, to discriminate PDAC, chronic pancreatitis, and normal pancreas [23]. We recently proposed the high expression of MUC4/MUC16/MUC20 signature as a marker of poor prognosis for pancreatic cancers [24].

Two additional atypical endothelial mucins (MUC14/EMCN and MUC18/MCAM) have been described in cancer [25,26]. MUC14, also known as endomucin (EMCN), is an endothelial sialomucin that inhibits endothelial–leukocyte interaction and has been reported to be a poor prognosis marker in gastric cancer [27]. MUC18 also called melanoma cell adhesion molecule (MCAM) or CD146, is a glycoprotein that is associated with gallbladder cancer and melanoma and that is also detected in pancreatic cancer stroma [28].

In the present manuscript, we used web-tools such as GEPIA, cBioPortal, SurvExpress, and PROGgeneV2 to investigate the expression pattern of every mucin gene in the pancreatic cancer PAAD dataset available from the TCGA consortium. We analyzed the impact of their expression on patient survival and the correlation of mucin gene expression, and we performed a hierarchical clustering analysis of PAAD cohort. We identified two major subgroups: one with a membrane-bound mucin signature (*MUC1, -4, -16, -17, -20,* and *-21*; signature #2) and the other one with an atypical mucin signature (*MUC15, MUC14/EMCN,* and *MUC18/MCAM*, signature #1). Signature #2 is associated with shorter overall survival and is proposed as a poor prognosis signature.

## 2. Results

### 2.1. Mucin Expression Patterns in Pancreatic Adenocarcinoma

In order to study mucin gene expression, we took advantage of publicly available datasets. We analyzed the expression of every mucin gene in pancreatic adenocarcinoma samples compared to normal pancreas tissues using GEPIA to compile Genome Tissue Expression (GTEX) (normal) and TCGA (tumor) datasets. We generated boxplots for genes encoding membrane-bound mucins (*MUC1, MUC3A, MUC4, MUC12, MUC13, MUC15, MUC16, MUC17, MUC20, MUC21* and *MUC22*) (Figure 1A), genes encoding secreted mucins (*MUC2, MUC5AC, MUC5B, MUC6, MUC7, MUC9/OVGP1*, and *MUC19*) (Figure 1B), and the two atypical mucin genes *MUC14*/*EMCN* and *MUC18*/*MCAM* (Figure 1C).

We observed a statistically significant increased expression in tumor samples for a majority of the genes encoding membrane-bound mucins (*MUC1, -3A, 4, -12, -13, -16, -17,* and *-20*) (*p <* 0.01) (Figure 1A). Only *MUC15, MUC21*, and *MUC22* mRNA did not harbor any significant expression variation. Two secreted mucin mRNA (*MUC5AC* and *MUC5B*) were also increased in tumor samples (*p <* 0.01), whereas *MUC6* was decreased in tumor samples compared to the normal pancreas (Figure 1B). The atypical mucins mRNA (*MUC14/EMCN* and *MUC18/MCAM*) were also increased in tumor samples (*p <* 0.01) (Figure 1C). Moreover, our analysis of the GSE28735 dataset, which contains 45 tumor samples and 45 paired adjacent nontumoral samples, showed a similar pattern with a significant increase in membrane-bound *MUC1, MUC3, MUC4, MUC13, MUC16, MUC17, MUC20*, and *MUC21* mucin mRNA (Appendix A) and the *MUC5B* secreted mucin mRNA (Appendix A). We observed a decrease in the expression of genes encoding membrane-bound *MUC15* (Appendix A) and secreted *MUC7* and *MUC19* in tumor samples (Appendix A) (*p* < 0.01). A decrease in *MUC6* expression was also close to statistical significance (*p =* 0.059). On the contrary, the increase in *MUC14*/*EMCN* and *MUC18*/*MCAM* was not confirmed in this dataset (Appendix A).

Altogether, we observed a dramatic change of the mucin gene expression pattern in PDAC notably characterized by a global increase in the genes encoding membrane-bound mucin burden.

### 2.2. Mucins and Patient Survival

We then searched for a possible association between the mucin gene expression and patient survival using the SurvExpress tool. We extracted hazard ratios (HR) from the PAAD-TCGA cohort using the optimized algorithm that splits patients into two cohorts where the *p*-value is minimal. We observed that poorer patient survival is strongly associated with the high expression of *MUC1* (hazard ratio (HR) = 4.49, confidence interval (CI) 1.64–12.33), *MUC4* (HR = 3.94, CI 1.81–8.61), and *MUC5B* (HR = 4.38, CI 1.76–10.9) (Figure 2). High expression of *MUC16* (HR = 2.53, *p =* 0.0008573), *MUC17* (HR = 1.92, *p =* 0.01448), *MUC20* (HR = 2.82, *p =* 0.01545), and *MUC21* (HR = 1.88, *p =* 0.01512) mRNA was also associated with worse overall survival. *MUC14*/*EMCN* atypical high expression was associated with lower survival (HR = 2.36, *p =* 0.03346).

We then conducted a similar analysis using PROGgeneV2 that splits the PAAD cohort into two cohorts on the basis of the median of expression. We confirmed a similar association between high expression of *MUC1*, *MUC4*, *MUC16*, *MUC17*, *MUC20*, and *MUC21* and shorter patient survival (Appendix A). In addition, high expression of *MUC2* and *MUC5B* was also associated with a worse prognosis in this analysis. On the contrary, *MUC9* high expression was correlated with a better outcome of PAAD patients. Finally, the analysis of GSE57495 containing 63 patients showed a trend toward an association of *MUC1*, *MUC4*, *MUC16*, *MUC17*, and *MUC20* with a worse outcome that was close to statistical significance (Appendix A).

### 2.3. Mucin Genomic Alterations in Pancreatic Adenocarcinoma

We extracted genomic alterations and RNA-Seq by Expectation-Maximization (RSEM) values for every mucin gene in the PAAD-TCGA cohort using quantification data from RNAseq available with the cBioPortal tool. We could not retrieve any RSEM values for *MUC3*, *MUC5AC*, *MUC8*, *MUC19*, and *MUC22*. For the following analyses, we focused our work on available mucin data. The oncoprint is presented in Figure 3. In total, 63% of PDAC samples (105/168) harbored at least one mucin gene alteration event (amplification, deletion, mRNA high, or mutation) with *MUC1*, *MUC17*, *MUC16*, and *MUC4* being the most frequently altered genes (15%, 15%, 14%, and 9%, respectively). There was no significant statistical difference in survival between altered and nonaltered groups (Appendix A). We performed a similar analysis for individual mucins. Only *MUC16* alterations were associated with worse overall survival and shorter progression-free survival (Appendix A). We also investigated the most frequent gene mutations in altered and nonaltered groups. Interestingly, we observed an enrichment of the cornerstone *V-Ki-ras2 Kirsten rat sarcoma viral oncogene homolog* (*KRAS*) mutation in the mucin gene-altered group (70.48% vs. 57.14%, *p =* 0.0564). Furthermore, *Titin* (*TTN*) (23.81% vs. 1.59%, *p* < 0.001), *Ras-responsive element-binding protein 1* (*RREB1*) (6.67% vs. 0%, *p =* 0.0345), and *cyclin-dependent kinase inhibitor 2A* (*CDKN2A*) (24.76% vs. 12.7%, *p =* 0.0435) were the most enriched gene mutations occurring in the mucin altered group (Appendix A).

### 2.4. Correlation among Mucin Gene Expression in Pancreatic Adenocarcinoma

We analyzed Pearson correlation coefficient for each combination of mucin genes and realized a principal component analysis (PCA). We identified 26 combinations of mucin genes with a positive correlation (0.17 < *r* < 0.85, *p <* 0.05) and 11 with a negative correlation (−0.46 < *r* < −0.17, *p <* 0.05) (Figure 4A and Table 1). Among positive correlations, we observed a very strong correlation among *MUC2*, *MUC12*, and *MUC13* (*r* = 0.72–0.85). Membrane-bound mucins mRNA (*MUC1*, *MUC4*, *MUC16*, *MUC17*, *MUC20*, and *MUC21*) were also frequently correlated suggesting a global membrane-bound mucin pattern. On the contrary, we observed that atypical mucin mRNAs (*MUC14*/*EMCN* and *MUC18*/*MCAM*) were correlated with each other (*r* = 0.44) but were negatively correlated with most membrane-bound mucins (*MUC1*, *MUC4*, *MUC13*, *MUC16*, *MUC17*, and *MUC20*; −0.17 < *r* < −0.44). Results of PCA confirmed that the membrane-bound mucin cluster was negatively correlated with the *EMCN*/*MCAM* cluster (Figure 4B). These two clusters appear to be independent of *MUC2*, *MUC5B*, *MUC12*, and *MUC13* expression as illustrated on the PCA plot (Figure 4B). Similar analysis was conducted using the independent Queensland Centre for Medical Genomics (QCMG) pancreatic cancer dataset containing 96 transcriptomes [4] (Appendix A). We observed a similar correlation of membrane-bound mucins MUC1, MUC4, MUC13, MUC16, MUC17, and MUC20 (0.42 < *r* < 0.64) (Appendix A) (Appendix A, Appendix A). *MUC14*/*EMCN* and *MUC18*/*MCAM* mRNAs were also correlated with each other (*r* = 0.34) (Appendix A).

### 2.5. Unsupervised Hierarchical Clustering of the PAAD Cohort

We used an unsupervised hierarchical clustering algorithm on the PAAD cohort data according to the relative mucin gene mRNA levels. The unsupervised hierarchical clustering analysis grouped patients with similar mucin characteristics. We obtained four groups of patient samples as illustrated on the dendrogram (Figure 5A).

The major clusters #1 (red) and #2 (yellow) contained 97 and 67 patients, respectively. We also obtained two small groups of patients that contained only one and three patients (#3 and #4, respectively). Therefore, we focused on clusters #1 and #2 for further analyses. Cluster #1 was characterized by a higher expression of *MUC14*/*EMCN, MUC15*, and *MUC18*/*MCAM* mucin mRNAs (Figure 5B). Cluster #2 was characterized by a global overexpression of genes encoding membrane-bound mucins (*MUC1, -4, -16, -17, -20,* and -*21*) (Figure 5B). All these differences in expression were statistically significant (*p <* 0.01). Interestingly, Kaplan–Meier curves showed that cluster #2 was associated with shorter overall survival than cluster #1 (*p =* 0.05, HR = 1.5, CI 1–2.34) (Figure 5C) but did not impact the progression-free survival (*p =* 0.15) (not shown). Similar hierarchical clustering analysis was realized using the PAAD cohort without the four patients belonging to clusters #3 and #4. We obtained four new clusters (not shown). In total, 139 patients out of 164 were sorted into corresponding groups (84% concordance). Cluster #2 was divided into two clusters characterized by two membrane-bound mucin signatures (*MUC1/13/17/20/12* or *MUC4/16/20/21*). An independent unsupervised hierarchical clustering algorithm on the QCMG dataset was also performed. We obtained three clusters as illustrated in Appendix A (Appendix A). Cluster #1′ was characterized by a higher expression of *MCAM*. Clusters #2′ and #3′ were more related and were characterized by a higher expression of membrane-bound mucins (*MUC1* and *MUC17* for cluster #2′; *MUC4*, *MUC16*, and *MUC20* for cluster #3′).

We then analyzed the clinical characteristics of patients belonging to clusters #1 and #2. We observed a trend toward male enrichment in the cluster #2 (*p =* 0.077) (Figure 6A). Age of diagnosis was equivalent in both groups (65 years old, Figure 6B). Additionally, we observed a significant increase in whole-genome gene mutation counts in patients from cluster #2 (*p <* 0.001) (Figure 6C). We did not obtain evidence for significant differences in clinical features (tumor stage, differentiation grade, lymph node invasion) (Figure 6D–F) suggesting that the mucin expression pattern could be an independent prognostic biomarker in PDAC.

We also investigated the survival of PDAC patients in relation to the relative expression levels of the two clusters using SurvExpress that splits the cohort where the *p*-value is minimal and PROGgeneV2 that separates the two cohorts depending on the median of expression of the genes of interest. We also investigated the GSE57495 independent dataset that contains relative mRNA levels and survival of 63 PDAC patients. Using the SurvExpress tool, we observed that signature #2 was associated with a higher hazard ratio (HR = 4.3) than signature #1 (HR = 1.75) in the PAAD-TCGA dataset (Figure 7A). Indeed, the high mRNA expression of *MUC1*, *MUC4*, *MUC16*, *MUC17*, *MUC20*, and *MUC21* was correlated with shorter overall survival (*p =* 0.000195) (Figure 7B). The high mRNA expression of *MUC14*/*EMCN*, *MUC15*, and *MUC18*/*MCAM* was also associated, to a lower extent, with shorter survival in the PAAD dataset (Figure 7C). Using PROGgeneV2, we confirmed that the high expression of signature #2 was associated with a worse outcome compared to signature #2 low-expression tumors in both TCGA-PAAD and GSE57495 (HR = 1.18–1.45, *p <* 0.001) (Appendix A). On the contrary, signature #1 did not reach statistical significance in the TCGA-PAAD (*p =* 0.055) or in the GSE57495 dataset (*p =* 0.43) (Appendix A). Altogether, this suggests that signature #2 could be useful in the stratification of patients with worse prognosis in pancreatic cancer.

### 2.6. Immunohistochemistry (IHC) Analysis of Mucin Expression in Human PDAC Samples

Lastly, in order to investigate the cellular distribution of the different mucin proteins in human PDAC samples, we retrieved IHC data from Protein Atlas that were performed on tissue macro-arrays containing 10 to 12 different PDAC samples. Regarding the membrane-bound signature (#2), we observed that 12/12 and 7/11 samples expressed MUC1 and MUC4 membrane-bound mucins, respectively (Figure 8A). We observed strong cytoplasmic and/or membrane staining (Figure 8B). Only 3/12 samples harbored MUC16 expression. MUC17 and MUC21 were not detected in any PDAC samples. MUC20 was not evaluated. Mucins that belong to signature #1 were also investigated. MUC14/EMCN, MUC15, and MUC18/MCAM were detected in 8/10, 4/12, and 5/11 PDAC samples, respectively. Interestingly, MUC14/EMCN, which is described as an endothelial mucin, was expressed by tumor cells (Figure 8B). On the contrary, MUC18/MCAM staining was only observed in the stroma surrounding the tumor cells.

## 3. Discussion

In the present study, we analyzed the PAAD cohort of the TCGA and identified a mucin signature that correlates with a worse outcome for patient survival, suggesting its potential usefulness as a prognostic marker. TCGA helps the scientific community to decipher the molecular landscape of PDAC and provides a roadmap for precision medicine [9]. PDAC is a dramatic disease with interpatient heterogeneity [8]; therefore, a better understanding of each tumor is mandatory for better healthcare and precision medicine.

We identified a patient cluster characterized by a higher expression of 6 membrane-bound mucin genes in the tumor cells. Membrane-bound mucins are commonly associated with increased tumor progression. Among them, MUC1 and MUC4 have been extensively demonstrated to be oncomucins promoting tumor aggressiveness, proliferation, and metastasis [1,29,30,31], suggesting the potential use of their expression and methylation status as independent prognosis markers [32,33,34]. MUC16 and MUC20 have also been implicated in the malignant phenotype (cell viability, migration, and invasion) of PDAC cells [18,35]. MUC17 has been previously proposed as a potential prognosis molecular marker of PDAC [36]. MUC21 was more recently discovered in lung adenocarcinoma [37] and has not been described in pancreatic pathologies. Interestingly, MUC1 and MUC4 may regulate the response to chemotherapy. Notably, MUC1 and MUC4 silencing promote gemcitabine sensitivity [38,39,40]. There are limited chemotherapy protocols (gemcitabine/S1, FOLFIRINOX, Nab-Paclitaxel) highlighting the importance of transcriptomic research to orientate patients toward the best individual protocol for each patient. We hypothesize that mucin expression combination (notably MUC1 and MUC4) could also serve as a prognostic biomarker to tailor chemotherapy.

Global alteration (amplification/deletion/increased mRNA) of mucin pattern was not associated with overall survival in our analysis, but we observed that the patients showing individual *MUC16* mRNA alteration had a worse outcome than the nonaltered patients. This was contradictory with previous work that showed that PDAC long-term survivors harbor neoantigens in the tumor antigen MUC16/CA125 [41]. The presence of neoantigens was proposed as a biomarker for immunogenic tumors. This is consistent with the finding that patients belonging to cluster #2 harbored a lower whole-genome mutation count. The alteration of neoantigen formation remains to be demonstrated between the two clusters. Additionally, we also observed an enrichment of *TTN*, *RREB1*, and *CDKN2A* gene mutation within the patient harboring at least one mucin gene alteration. *TTN* has been previously shown to be independently correlated with the overall survival of PDAC patients. *TTN* expression is significantly lower in PDAC cell lines compared to human pancreatic ductal epithelial (HPDE) cells [42]. On the contrary, *RREB1* is upregulated in PDAC tissues and regulates the proliferation and migration of PDAC cells [43]. *CDKN2A* that encodes the tumor suppressor p16/INK4 is well known as *CDKN2A* germline mutations have been reported in familial forms of pancreatic cancer [44]. The meaning of the correlation between mucin gene alteration and these three gene mutations remains to be fully understood.

Most mucins are epithelial markers, but MUC14, also classified as a mucin because of its *O*-glycosylated structure, is designated as an endomucin (EMCN) and is a biomarker of endothelial cells [44]. MUC18/MCAM/CD146 is expressed in cancer-associated fibroblasts that are enriched in PDAC stroma [28]. We observed for the first time a patient subgroup that is characterized by a higher expression of *MUC14*/*EMCN*, *MUC18*/*MCAM*, and *MUC15* in their tumor tissue. According to Moffitt’s previous work, two stromal subtypes (normal and activated) were defined and harbor independent prognostic characteristics [6]. We hypothesize that the *MUC14/MUC18/MUC15* subgroup could overlap with the different stromal populations. Recent work also showed that the expression levels of *MUC14/EMCN*, *MUC18/MCAM*, and *MUC15* individually showed significant correlations with worse overall survival in stomach adenocarcinoma (STAD) from TCGA dataset [27] suggesting that these genes could also be prognostic biomarkers in other digestive cancers. However, this could be specific to gastric cancer since we observed that MUC14 and MUC18 are associated with better survival in the PDAC cohort. MUC5AC and MUC6 gastric-type mucin expressions are detected in PDAC and their neoplastic lesions [16]. MUC5AC is also expressed in other types of pancreatic lesions such as the intestinal type of intraductal papillary mucinous neoplasm (IPMN) [45]. Previous work showed that MUC5AC could be a therapeutic target in PDAC (using NPC-1 or clivatuzumab/PAM4 antibodies) and a potential biomarker (as circulating marker and combined with CA19.9) [23,46]. However, in the present analysis, the expression of these two mucins was not associated with the patient clustering. MUC5B has been shown to regulate migration and survival in AsPC-1 PDAC cells [47]. Interestingly, we observed that MUC5B is aberrantly expressed in tumor samples and that MUC5B high expression is associated with poorer survival. Globally, all the genes encoding secreted mucins appeared to be grouped together but were independent regarding the two mucin signatures (as illustrated on Figure 4B).

A recent unsupervised clustering analysis as a function of *MUC1*, *MUC2*, and *MUC4* identified a patient cluster, characterized by a higher expression level of the three mucin genes harboring a significantly poorer prognosis. Further machine learning analysis showed that the methylation status of these three mucin genes could be potential prognostic biomarkers [48]. The correlation between mucin gene expression and the two identified signatures suggests a concerted regulation of these mucin genes as previously suggested for secreted mucins [13,49]. Among possible cellular mechanisms driving concerted regulation, hyper or hypomethylation of the membrane-bound mucins could explain the overall increased expression of these genes [50]. *MUC1* and *MUC4* hypomethylation status were shown to be correlated with expression of DNA demethylation factors (*TET1*, *TET2*) and *DNMT3* DNA methyltransferase [33]. We also observed a nearly significant enrichment of KRAS mutation in patients harboring at least one mucin gene alteration. Interestingly, *MUC4* and *MUC16* were shown to be direct target genes of K-ras [51,52]. Further analyses are necessary regarding *MUC17*, *MUC20*, and *MUC21* regulation by K-ras or methylation to confirm this hypothesis. It was also suggested that mucin mRNAs share common mechanisms of stabilization that may participate to their singularity as hyperstable transcripts [53] and to their co-expression. In addition, this unusually long half-life supports the relevance of using transcriptomics data to unravel mucin signatures of cancers of different subtypes.

In addition to the interpatient heterogeneity that we observed in the present manuscript, intratumor heterogeneity (ITH) within a single tumor is characteristic of PDAC. Genomic and epigenomic diversity of PDAC tumor cells leads to diversity of transcriptome and proteome between tumor PDAC subclones [21]. Desmoplasia is a major feature in PDAC with an important extracellular matrix (ECM) deposition surrounding tumor cells and high content of cancer-associated fibroblasts and immune cells. This ITH should be taken into account and suggests that the proposed mucin signature should be validated by IHC in order to decipher the cell type involved in the different mucin expression. Despite being described as an endothelial marker, we mainly observed a MUC14/EMCN staining in tumor cells. On the contrary, MUC18/MCAM staining was mainly localized in the stroma. These surprising patterns will have to be confirmed and compared to the grade of desmoplasia. However, we hypothesize that the higher expression of CAF-MUC18 in some PDAC patients could be correlated with the patients showing the highest grade of desmoplasia.

In the near future, we propose to explore the extent and clinical implications of ITH and decipher the mucin signature between patients and tumor regions and discuss how profiling the mucin expression pattern could help clinicians for precision medicine, lead to new prognostic signatures, and notably participate in assigning the best therapy regimen for each patient.

## 4. Materials and Methods

### 4.1. Expression Data Retrieval and Analysis

MUC1, MUC2, MUC3A, MUC4, MUC5AC, MUC5B, MUC6, MUC8, MUC9, MUC12, MUC13, MUC14/EMCN, MUC15, MUC16, MUC17, MUC18/MCAM, MUC19, MUC20, and MUC21 expression was extracted from GEPIA [54]. The normal (n = 171) vs. tumor (n = 179) patterns of expression were analyzed using GEPIA that recomputes the Genome Tissue Expression (GTEX) and TCGA gene expression data.

The GSE28735 microarray was analyzed using the National Center for Biotechnology Information (NCBI) Gene Expression Omnibus (GEO) database (http://www.ncbi.nml.nih.gov/geo/). GSE28735 is a dataset containing 45 normal pancreas (adjacent non tumoral, ANT) and 45 tumor (T) tissues from PDAC cases. The mRNA expression in tumor and ANT samples was analyzed using R studio.

All queries for each mucin gene were also realized in pancreatic adenocarcinoma PAAD dataset from TCGA (http://cancergenome.nih.gov/)) and QCMG [4] using the cBioPortal website [55,56]. The mRNA expression values were retrieved as RSEM (batch-normalized from Illumina HiSeq_RNASeqV2). Gene expression and clinical information were filtered for missing values.

All TCGA data analyses were processed using R studio. The complete R script is available at https://github.com/NicolasJonckheere/Mucin-script. The Pearson correlation coefficients (*r*-values) and *p*-values were calculated for each combination of mucin genes. Principal component analysis (PCA) and unsupervised hierarchical clustering (HC) were performed using the FactoMiner package in R studio. The number of clusters is automatically determined by HCPC {FactoMineR}. The default partition in four clusters is the one with the higher relative loss of inertia. Survival analyses of patient clusters were conducted using the survival and SurvMiner packages. Only major clusters #1 and #2 were further analyzed. Clusters #3 and #4 were excluded from the following analyses due to a low number of cases (one and three patients, respectively).

### 4.2. Survival Analysis

Survival analysis was conducted using the SurvExpress and PROGgeneV2 online tools [57,58]. Individual mucin gene analysis and the signature combination analysis were performed using SurvExpress with the optimized Maximize algorithm that attributes a minimal *p-*value to a risk group. Hazard ratio (95% confidence interval (CI)) was also evaluated. Combinations of mucins (#1: *MUC15*, *EMCN*, and *MCAM*; #2: *MUC1*, *MUC4*, *MUC16*, *MUC17*, *MUC20*, and *MUC21*) were also analyzed using PROGgeneV2 in TCGA-PAAD and GSE57495 datasets. GSE57495 is an independent microarray analysis of 63 patients with pancreatic cancer tissues used to predict overall survival [59].

### 4.3. Immunohistochemistry (IHC)

Mucin expression was analyzed by IHC using the available scanned tumor macro array staining from ProteinAtlas database (www.proteinatlas.org/) [36]. In total, 10 to 12 PDAC samples were analyzed for MUC1 (HPA004179), MUC2 (CAB016275), MUC3A (HPA010871), MUC4 (HPA005895), MUC5AC (HPA040615), MUC5B (HPA008246), MUC6 (CAB002165), MUC7 (HPA006411), MUC9 (HPA062205), MUC12 (HPA023835), MUC13 (HPA045163), MUC14 (HPA005928), MUC15 (HPA026110), MUC16 (CAB055172), MUC17 (HPA031634), MUC18 (HPA008848), and MUC21 (HPA052028). No information could be retrieved for MUC19, MUC20, and MUC22. IHC staining was automatically determined and eventually manually adjusted by Protein Atlas experts and was indicated as high/medium/low staining or not detected.

### 4.4. Statistical Analysis

Chi-square, ANOVA, and Student *t*-test statistical analyses were performed using R studio (https://rstudio.com/) and Graphpad Prism 6.0 software (Graphpad softwares Inc., La Jolla, CA, USA). A *p*-value *<*0.05 was considered statistically significant. The GEPIA tool also provided *t*-test analysis. SurvExpress and PROGgeneV2 tools provided statistical analyses of hazard ratios and overall survival. A log-rank test was used to evaluate the equality of survival curves between the high- and low-risk groups.

## 5. Conclusions

In the present work, we investigated available PDAC datasets combining mRNA expression and patient clinical features. We observed that PDAC is globally associated with an increase in the mRNA relative level of most membrane-bound mucins. We performed an unsupervised hierarchical clustering analysis that allowed us to identify a mucin signature, characterized by an increased *MUC1/4/16/17/20/21* mRNA level that correlates with a worse outcome for patient survival, suggesting its potential usefulness as a prognostic marker. We hypothesize that a better knowledge of the mucin expression pattern could help clinicians for precision medicine.

## Figures and Tables

**Figure 1 cancers-12-03309-f001:**
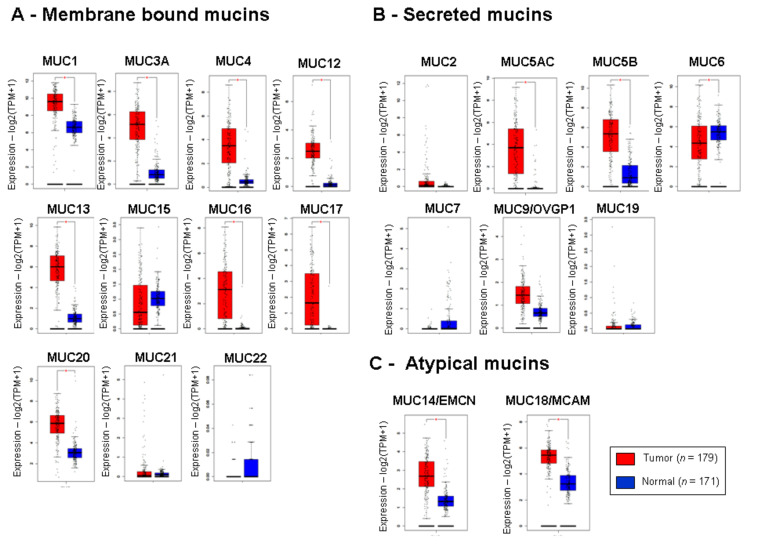
Relative mucin messenger RNA (mRNA) levels in pancreatic adenocarcinoma and normal pancreas tissues. The whisker boxplots for the mRNA of membrane-bound mucins (*MUC1, MUC3, MUC4, MUC12, MUC13, MUC15, MUC16, MUC17, MUC20, MUC2*1, and *MUC22*) (**A**), secreted mucins (*MUC2, MUC5AC, MUC5B, MUC6, MUC7, MUC8, MUC9/OVGP1,* and *MUC19*) (**B**), or atypical mucins (*MUC14/EMCN, MUC18/MCAM*) (**C**) were generated using GEPIA in The Cancer Genome Atlas (TCGA) and Genome Tissue Expression (GTEX) samples. Mucin mRNA relative levels are expressed as log_2_ transcripts per million bases (TPM). Statistical analyses were performed using an unpaired *t*-test (* = *p <* 0.05).

**Figure 2 cancers-12-03309-f002:**
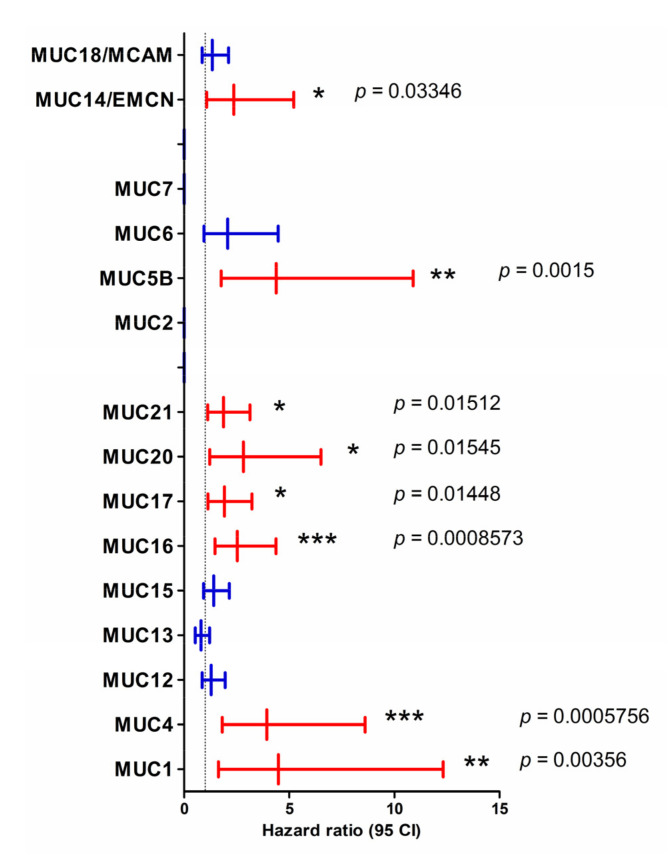
Hazard ratio in high and low expression levels of mucin genes in pancreatic adenocarcinoma (PAAD) cohort. Hazard ratio (HR) was calculated in population designated as high risk and low risk (higher value of mucin gene of interest for higher risk) on the basis of individual mucin expression by the SurvExpress optimized algorithm in PAAD from TCGA datasets. Statistically significant hazard ratios (*p <* 0.05) are represented in red. The *p*-value is indicated on the graph (***, *p <* 0.001; **, *p <* 0.01; *, *p <* 0.05).

**Figure 3 cancers-12-03309-f003:**
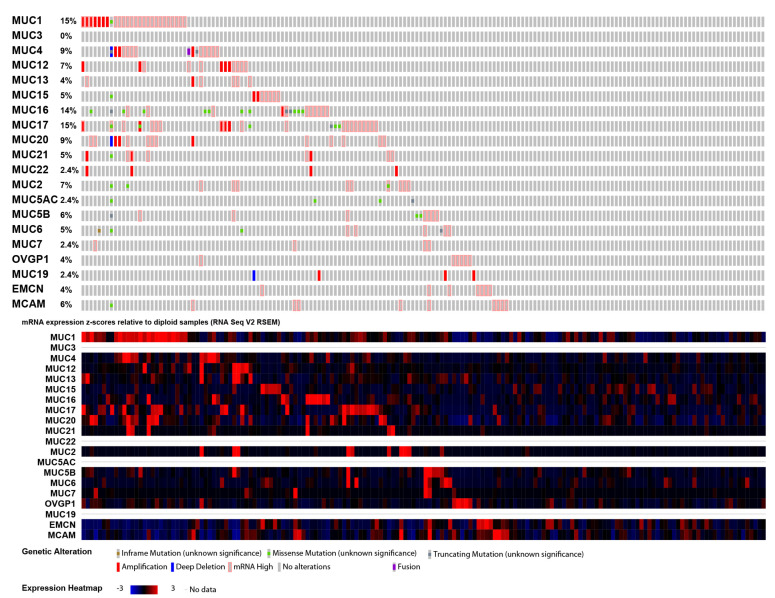
cBioPortal oncoprint of mucin genes in pancreatic adenocarcinoma PAAD cohort of the TCGA dataset. Mucin genes data were extracted using cBioPortal to illustrate every mucin gene alteration (gene amplification, deletion, mutations, and mRNA high levels) for each patient of the PAAD cohort. The heatmap illustrates the mRNA level for each patient.

**Figure 4 cancers-12-03309-f004:**
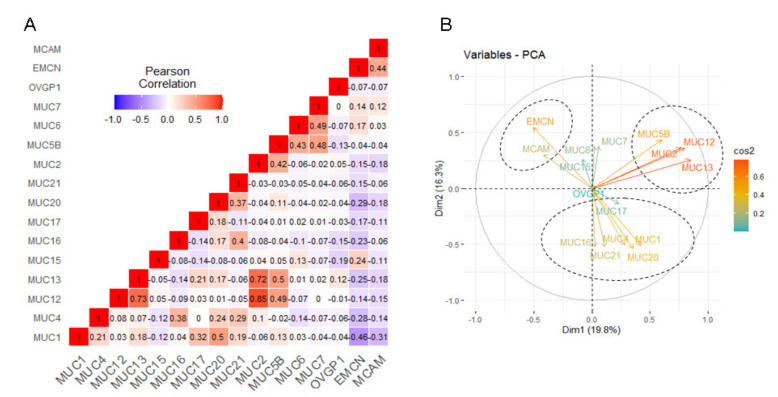
Correlation analysis of relative mucin mRNA levels in pancreatic adenocarcinoma (PAAD) cohort of the TCGA. (**A**) Correlation Pearson *r* values were calculated for each mucin mRNA combination from PAAD cohort. (**B**) Principal component analysis (PCA) of mucin mRNA relative expression in PAAD cohort. Positively correlated variables are grouped together, whereas negatively correlated variables are on opposite sides. A 90° angle, formed by two arrows, illustrates independency of variables. Cos^2^ (square cosine, squared coordinates) indicates the quality of representation of the variables on factor map.

**Figure 5 cancers-12-03309-f005:**
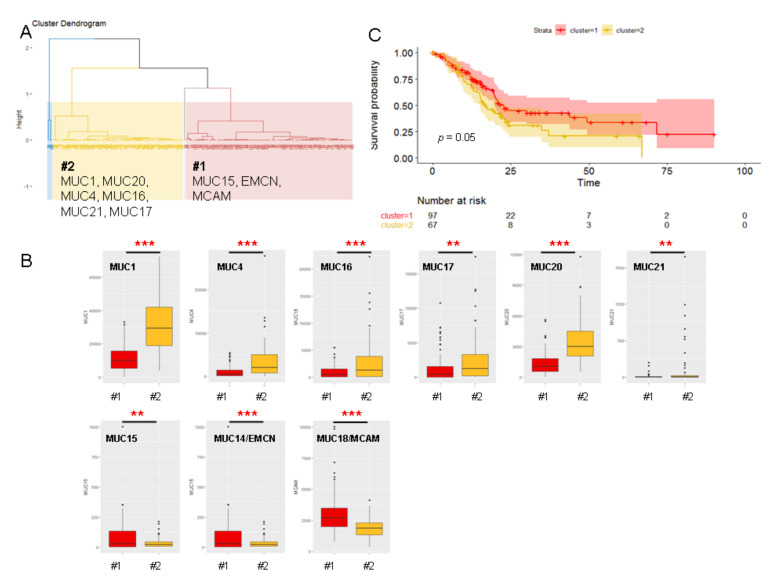
Unsupervised hierarchical clustering analysis of PDAC patients according to their relative mucin mRNA expression. (**A**) Mucin-based dendrogram that displays clustering analysis between patients. The dendrogram shows four hierarchical clusters according to the expression pattern of mucins. (**B**) Boxplot of mucin expression in major clusters #1 and #2. Statistical analyses were performed using an unpaired *t*-test (***, *p <* 0.001; **, *p <* 0.01). (**C**) Kaplan–Meier curves of clusters #1 and #2.

**Figure 6 cancers-12-03309-f006:**
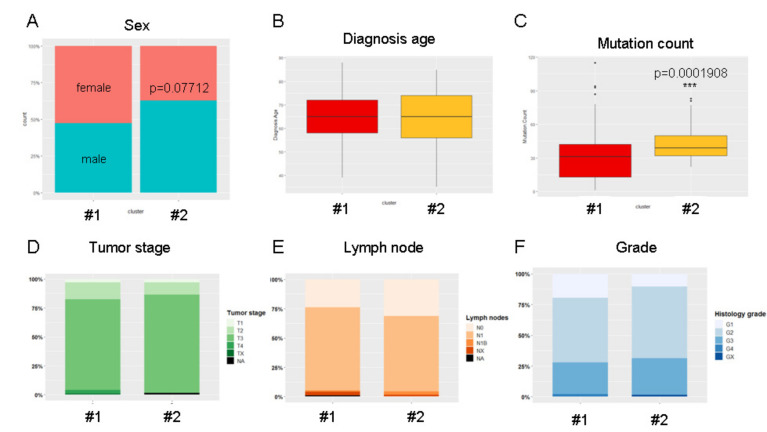
The mucin-based stratification is independent of clinical features and is associated with the mutation count in PDAC patients. (**A**) Percentage-staked bar plot of sex distribution in PAAD-TCGA cohort of clusters #1 and #2. Boxplots showing age of diagnosis (**B**) and mutation counts (**C**) for clusters #1 and #2 (***, *p <* 0.001). Percentage-staked bar plot showing tumor stage (T1/T2/T3/T4/TX/NA) (**D**), lymph node status (N0/N1/N1B/NX/NA) (**E**), and differentiation grades (G1/G2/G3/G4/GX) (**F**) in clusters #1 and #2.

**Figure 7 cancers-12-03309-f007:**
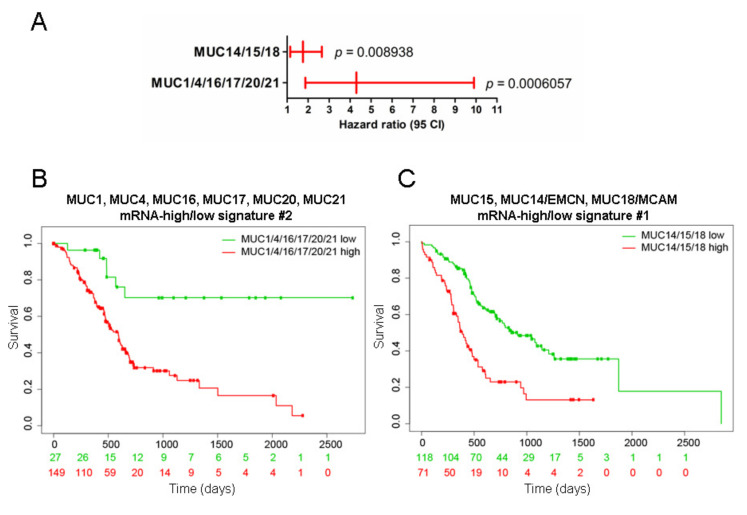
Analysis of overall survival of the #1 and #2 mucin signatures in pancreatic cancer datasets using SurvExpress. (**A**) Hazard ratio according to signature #1 (*MUC1*, *MUC4*, *MUC16*, *MUC17*, *MUC20*, and *MUC21*) and #2 (*MUC14*/*EMCN*, *MUC15*, and *MUC18*) high and low expression in the pancreatic adenocarcinoma PAAD cohort. (**B**) PAAD patients were stratified using a gene signature combining *MUC1, MUC4*, *MUC16*, *MUC17*, *MUC20*, and *MUC21*. Kaplan–Meier curves were analyzed using the optimized SurvExpress Maximize algorithm. The number of analyzed patients across time (days) is indicated below the horizontal axis for both conditions. (**C**) PAAD patients were stratified using a gene signature combining *MUC15*, *MUC14*/*EMCN*, and *MUC18*/*MCAM* using the Maximize algorithm.

**Figure 8 cancers-12-03309-f008:**
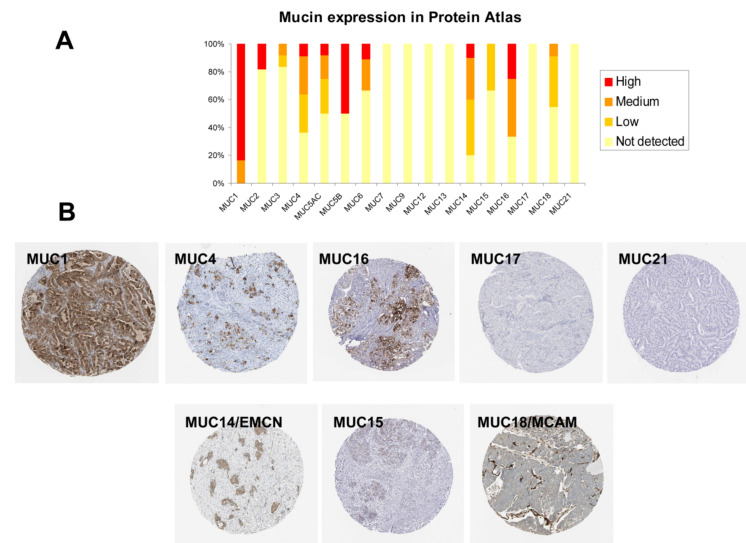
Detection of mucin expression by immunochemistry in human pancreatic cancer human tissue. (**A**) Histogram of mucin expression in PDAC samples from Protein Atlas (www.proteinatlas.org/). In total, 11–12 samples were analyzed for MUC1, MUC2, MUC3A, MUC4, MUC5AC, MUC5B, MUC6, MUC7, MUC9, MUC12, MUC13, MUC14, MUC15, MUC16, MUC17, MUC18, and MUC21. Immunohistochemistry (IHC) staining was evaluated as high/medium/low staining or not detected. No information could be retrieved for MUC19, MUC20, and MUC22. (**B**) Representative IHC stainings for MUC1, MUC4, MUC16, MUC17, and MUC21 (signature #2) and MUC14/EMCN, MUC15, and MUC18 (signature #1). All mucins showed a membrane and/or cytoplasmic staining in tumor cells except MUC18/EMCN, which was detected in the stroma.

**Table 1 cancers-12-03309-t001:** Correlation of Mucin gene expression in PAAD-TCGA (*p* < 0.05).

Positive Correlation			
Gene A	Gene B	Pearson *r*	*p*-Value
*MUC12*	*MUC2*	0.85	<0.00001
*MUC12*	*MUC13*	0.73	<0.00001
*MUC13*	*MUC2*	0.72	<0.00001
*MUC1*	*MUC20*	0.50	<0.00001
*MUC13*	*MUC5B*	0.50	<0.00001
*MUC12*	*MUC5B*	0.49	<0.00001
*MUC6*	*MUC7*	0.49	<0.00001
*MUC5B*	*MUC7*	0.48	<0.00001
*EMCN*	*MCAM*	0.44	<0.00001
*MUC5B*	*MUC6*	0.43	<0.00001
*MUC2*	*MUC5B*	0.42	<0.00001
*MUC16*	*MUC21*	0.40	<0.00001
*MUC4*	*MUC16*	0.38	<0.00001
*MUC20*	*MUC21*	0.37	<0.00001
*MUC1*	*MUC17*	0.32	0.00003
*MUC4*	*MUC21*	0.29	0.00016
*MUC15*	*EMCN*	0.24	0.00170
*MUC4*	*MUC20*	0.24	0.00202
*MUC13*	*MUC17*	0.21	0.00579
*MUC1*	*MUC4*	0.21	0.00623
*MUC1*	*MUC21*	0.19	0.01578
*MUC1*	*MUC13*	0.18	0.01768
*MUC17*	*MUC20*	0.18	0.02016
*MUC16*	*MUC20*	0.17	0.02416
*MUC13*	*MUC20*	0.17	0.02598
*MUC6*	*EMCN*	0.17	0.02798
**Negative correlation**			
*MUC1*	*EMCN*	−0.46	<0.00001
*MUC1*	*MCAM*	−0.31	0.00004
*MUC20*	*EMCN*	−0.29	0.00018
*MUC4*	*EMCN*	−0.28	0.00029
*MUC13*	*EMCN*	−0.25	0.00087
*MUC16*	*EMCN*	−0.23	0.00272
*OVGP1*	*MUC15*	−0.19	0.01240
*MUC13*	*MCAM*	−0.18	0.01819
*MUC2*	*MCAM*	−0.18	0.01860
*MUC20*	*MCAM*	−0.18	0.01888
*MUC17*	*EMCN*	−0.17	0.02811

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
