# Peer review of "Unsupervised Hierarchical Clustering of Pancreatic Adenocarcinoma Dataset from TCGA Defines a Mucin Expression Profile that Impacts Overall Survival"

_cancers, 2020, doi:10.3390/cancers12113309_

Round 1
Reviewer 1 Report
- In introduction, Reference no. 2 that mentions about “short survival and 5-year survival rate” is very old. It should be replaced with some latest statistics and references.
- In introduction, authors have mentioned that “CA15-3 or CA19.9 are potential PDAC biomarkers but with a limited specificity or sensitivity”. Authors should also cite the studies where CA19.9 has been combined with other markers such as MUC5AC (PUBMED Id: 27845339) to improve the overall sensitivity and specificity as PDAC early diagnostic markers.
- In figure 1, the x-axis labeling is barely visible. The font size in all the figures should be increased so that figures are properly visible.
- In Discussion section, discuss in detail, the findings on secretory mucins MUC5AC and MUC5B in PDAC and about the information already known in literature eg. (PUBMED Id: 29415129).
- Due to mention of MUC5AC in figure legend, the Supplementary figure 1A should include the bar graphs for MUC5AC.
Author Response
Please find enclosed the R2 revised manuscript “Unsupervised Hierarchical Clustering of Pancreatic Adenocarcinoma Dataset from TCGA Defines a Mucin Expression Profile that Impacts Overall Survival”.
We thank the reviewers for their encouraging and constructing comments. The modifications in the body of the manuscript appear in red. We hope that you will find this revised version suitable for final publication.
Reviewer 1:
In introduction, Reference no. 2 that mentions about “short survival and 5-year survival rate” is very old. It should be replaced with some latest statistics and references.
We added the following reference with updated statistics
Siegel, R.L.; Miller, K.D.; Jemal, A. Cancer statistics, 2020. CA Cancer J Clin 2020, 70, 7-30.
In introduction, authors have mentioned that “CA15-3 or CA19.9 are potential PDAC biomarkers but with a limited specificity or sensitivity”. Authors should also cite the studies where CA19.9 has been combined with other markers such as MUC5AC (PUBMED Id: 27845339) to improve the overall sensitivity and specificity as PDAC early diagnostic markers.
The reference (Kaur et al., 2017) has been added in the introduction to highlight the potential of MUC5AC-CA19.9 combination as biomarkers.
Circulating MUC5AC has also been proposed as potential biomarker, alone or in combination with CA19-9, to discriminate PDAC, chronic pancreatitis and normal pancreas [23].
In figure 1, the x-axis labeling is barely visible. The font size in all the figures should be increased so that figures are properly visible.
Figure1 and Figure6 have been modified to improve the visibility of axis. Figure3 could not be modified but the resolution is high enough to read the text.
In Discussion section, discuss in detail, the findings on secretory mucins MUC5AC and MUC5B in PDAC and about the information already known in literature eg. (PUBMED Id: 29415129).
As suggested by the reviewer, a small paragraph regarding MUC5AC and MUC5B was added in the discussion section.
[MUC5AC is also expressed in another type of pancreatic lesions such as intestinal type of intraductal papillary mucinous neoplasm (IPMN) [45]. Previous works showed that MUC5AC could be a therapeutic target in PDAC (using NPC-1 or PAM4 antibodies) and also a potential biomarker (as circulating marker and combined with CA19.9) [23,46]…
… MUC5B has been shown to regulate migration and survival in AsPC-1 PDAC cells [47]. Interestingly, we observed that MUC5B is aberrantly expressed in tumor samples and that MUC5B high expression is associated with poorer survival.
Due to mention of MUC5AC in figure legend, the Supplementary figure 1A should include the bar graphs for MUC5AC.
We could not retrieve any data regarding MUC5AC in GSE28735. The legend in supplemental figure 1 has been modified.

Reviewer 2 Report
This paper investigates the expression pattern of mucin gene in pancreatic cancer datasets towards statistical exploratory (PCA), clustering, tests and modeling (survival analysis) tools. The paper is well written.
The statistical analyses of the data of interest are adapted and carried out with rigor, the results are well commented.
I propose to check if the conditions of the paired t-test used are verified by the data (for instance normality condition, even if with the sample size larger than 30, gaussian approximation may be considered).
The authors should also discuss and motivate the choice of the number of clusters (4) chosen in hierarchical clustering (Section 2.5).
May the authors run again the clustering algorithm without the patients in the two small groups (#3 and #4), see page 8.
Author Response
Please find enclosed the R2 revised manuscript “Unsupervised Hierarchical Clustering of Pancreatic Adenocarcinoma Dataset from TCGA Defines a Mucin Expression Profile that Impacts Overall Survival”.
We thank the reviewers for their encouraging and constructing comments. The modifications in the body of the manuscript appear in red. We hope that you will find this revised version suitable for final publication.
Reviewer 2:
I propose to check if the conditions of the paired t-test used are verified by the data (for instance normality condition, even if with the sample size larger than 30, gaussian approximation may be considered).
Thanks to the reviewer, we corrected a mistake. Figure 1 statistical analysis was performed using unpaired t test. The 178 tumour and 171 normal samples were not paired. Unfortunately, we cannot verify the conditions ourselves since graphs were generated using GEPIA that do not provide raw data.
On the contrary, supplemental figure 1 analysis was performed using paired t test. As the reviewer noted, we analyzed 45 paired tumor and normal samples. We also verified the distribution that showed Gaussian distribution for every mucins as shown below.
The authors should also discuss and motivate the choice of the number of clusters (4) chosen in hierarchical clustering (Section 2.5).
This was added to the material and methods sections.
The number of clusters is automatically determined by HCPC {FactoMineR}. HCPC performs a hierarchical clustering from a PCA analysis. The suggested partition in 4 clusters is the one with the highest relative loss of inertia (automatically determined by HCPC). Moreover, the shape of the dendrogram confirmed the partitioning into 4 obvious clusters which allows visualisation of the global mucin signatures.
May the authors run again the clustering algorithm without the patients in the two small groups (#3 and #4), see page 8.
As requested, we performed a similar hierarchical clustering analysis in the PAAD cohort without the 4 patients belonging to clusters #3 and #4.
We obtained 4 new clusters (called 1b, 2b, 3b, 4b). 139 patients out of 164 were sorted in corresponding groups (84% concordance).
73 patients of cluster #1b were also in the cluster #1. The cluster #2 is divided in two clusters #3b and #4b that are characterized by two membrane bound mucins signatures.
53 and 13 patients from cluster #2 belonged in the cluster #3b and 4b, respectively.
